# A Comparative Analysis of Technical Efficiency and Profitability of Agribusiness and Non-Agribusiness Enterprises in Eastern DRC

**Dieu-Merci Akonkwa Nyamuhirwa [1,2,*], Bola Amoke Awotide [3], Doux Baraka Kusinza [4], Valery Kasereka Bishikwabo [2], Jacob Mignouna [5], Zoumana Bamba [6] and Paul-Martin Dontsop Nguezet [2]**

1 Faculty of Economics and Management, Université Catholique de Bukavu (UCB), Bukavu P.O. Box 285, Democratic Republic of the Congo

2 Social Science and Agribusiness, International Institute of Tropical Agriculture (IITA), Kalemie P.O. Box 570, Democratic Republic of the Congo; k.bishikwabo@cgiar.org (V.K.B.); p.dontsop@cgiar.org (P.-M.D.N.)

3 Monitoring, Evaluation and Learning, International Institute of Tropical Agriculture (IITA), Bamako P.O. Box 320, Mali; b.awotide@cgiar.org

4 Faculty of Economics, Social Sciences and Business Administration, Université de Namur, B-5000 Namur, Belgium; doux.barakakusinza@unamur.be

5 Olusegun Obasanjo Research Campus, International Institute of Tropical Agriculture (IITA), Bukavu P.O. Box 1222, Democratic Republic of the Congo; j.mignouna@cgiar.org

6 Capacity Development Unit, International Institute of Tropical Agriculture (IITA), Kinshasa P.O. Box 4163, Democratic Republic of the Congo; z.bamba@cgiar.org

* Correspondence: akonkwa.nyamuhirwa@ucbukavu.ac.cd

**Abstract:** The purpose of this study was to determine whether agribusiness could be competitive compared to non-agribusiness employment opportunities in terms of technical efficiency and profitability. We used data collected on all seven operating cassava community processing centers (CCPCs) and 150 comparable non-agribusiness enterprises in South Kivu province. A Data Envelopment Analysis (DEA), as well as cost–benefit ratios and net monthly revenue, were used to examine technical efficiency and profitability. Our results showed that agribusiness was more competitive than non-agribusiness in terms of technical efficiency and profitability. The cost–benefit ratio shows that every dollar invested in agribusiness earns investors US $2.8, while it earns investors in non-agribusiness US $2.1. Moreover, technical efficiency increases significantly with agribusiness. These results show that agribusiness can compete with other non-agribusiness activities, and it remains a solution to youth unemployment in the region.

**Keywords:** agribusiness; technical efficiency; Data Envelopment Analysis; economic profitability; eastern DRC

## 1. Introduction

Recent demographic trends show that the Democratic Republic of Congo (DRC) is one of the youngest countries in Africa [1–3]. For example, in 2019, 32% of the 87 million citizens were between the ages of 15 and 34, and the number of young people is expected to increase to 32 million by 2025 and 43 million by 2050 [4–6]. However, despite the increase in the number of higher education institutions from 764 in 2014 to 901 in 2015, the skills of young people do not meet the requirements of employers [7,8]. Therefore, unemployment is the biggest challenge for the DRC government, as it is for many African countries. The national unemployment rate is 65%, and that of youth is about 22.8% [9–12].

Given the potential of the DRC's agricultural sector, donors, governments, and development agencies have placed great importance on agribusiness as a means to reduce youth unemployment [13–22]. However, the agricultural sector in the DRC faces many challenges, of which inadequate access to land and market, low quality of infrastructure, low price competitiveness, an unattractive regulatory environment, and instability are on the top

of the list [20,23–27]. For example, the prices of the most consumed goods in the cities of Bukavu and Goma are on average 21% and 24% higher, respectively, than in neighboring Rwandan districts, such as Kamembe and Gisenyi, and it takes longer and costs more to start a business in the DRC than in neighboring countries, such as Rwanda [28–30]. In addition, food products from Rwanda sold in Bukavu markets have been found to make a high profit compared to products produced in the DRC [31].

Given the foregoing, agribusiness will be less attractive to youth if it does not enable them to earn sufficient income as with comparable activities outside agribusiness, such as stores and magazines, hairdressers, restaurants and bistros, pharmacies, and sewing workshops [32–34]. Nevertheless, to our knowledge, the existing empirical literature is sparse when it comes to comparing agribusiness and non-agribusiness enterprises and ensuring that agribusiness is more competitive than non-agribusiness enterprises [15,32,33]. A recent study [22] showed that in addition to the success of youth engagement interventions in agribusiness, there is a lack of evidence on what works. This study helps fill this important gap in the literature on agribusiness and sheds light on the employment potential of agribusiness in South Kivu by comparing the technical efficiency and profitability of agribusiness and non-agribusiness enterprises.

After Section 1 of the introduction, the rest of this study is organized as follows. Section 2 presents a brief description of the cassava community processing center (CCPC) model. Section 3 presents the methodology. Section 4 presents and discusses the findings, and Section 5 concludes and formulates recommendations.

## 2. Description of the Cassava Community Processing Center Model

Cassava remains a dominant tropical root crop and a source of food energy for more than 500 million people in Africa [21,35–37]. In the DRC, it contributes to nearly 60% of daily food energy intake per person, followed by maize [38–40]. However, national agricultural production has declined by about 60% since 1960 due to political instability, crop diseases, and inconsistent agricultural policies, resulting in a shortage of cassava supply [23,25–27,38]. Given the predominant role of cassava in Congolese daily consumption, international and national efforts are being pooled to strengthen the cassava value chain and ensure that a rapid crowding-in effect occurs. The development of the CCPCs model by the International Institute of Tropical Agriculture (IITA) is part of this effort. This rural enterprise model has several advantages (see Figure 1).

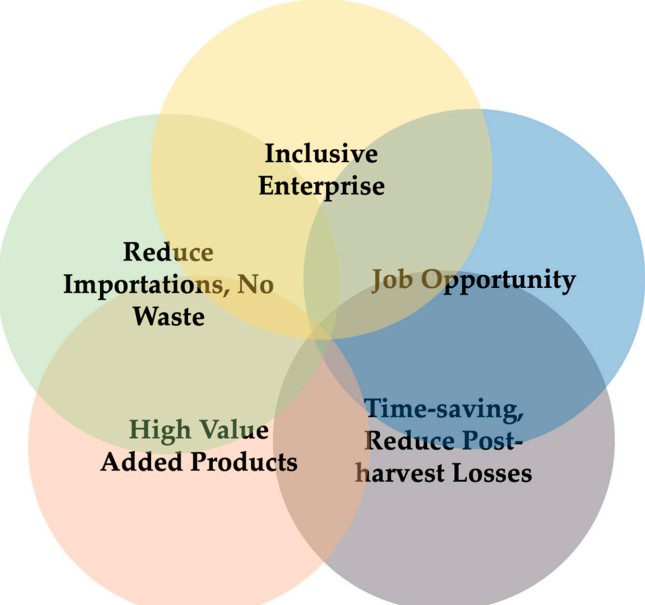

**Figure 1.** Five benefits of CCPC. Source. Authors' conception.

Specifically, CCPCs are based on and managed by local community organizations such as churches, farmer leader groups, farmer cooperatives, women's groups, and civil society. They promote both social cohesion and collective well-being, so first, they are an inclusive enterprise model. Second, CCPCs are suitable for young women and men as full-time employment because they are more labor-intensive. For example, peeling and washing cassava roots requires about 30 people, so the daily threshold of seven tons of cassava flour per processing can be reached (at a ratio of one ton of cassava roots in about 2.5 to 4 h). Third, the CCPC model has the advantage of requiring minimal treatment of the fresh root, which significantly reduces the time required for preparation, drying, and processing. The time saved significantly reduces post-harvest losses of cassava and allows farmers to devote time to other activities. Fourth, CCPCs produce a very high quality (VHQ) cassava flour, which is adequate for about 10 derivatives, such as gari, fufu, chips, cakes, bread, cookies, etc., thus enabling high-value addition along the cassava value chain. Finally, their development significantly reduces imports of cassava flour, as a CCPC can produce at least seven tons of cassava flour per day. This lowers the price for some period, increases the economic competitiveness of local cassava products, and gives consumers access to very high-quality cassava products that have a high nutritional value. In addition, from an environmental perspective, all residues are converted into animal feed, starch, or biofuels. See [35,36,41–43] for extensive literature on the benefits of cassava products.

This model has been implemented through various interventions across Africa, and in the Democratic Republic of Congo in particular, including the Support to Agricultural Research for Development of Strategic Crops in Africa (SARD-SC) project, the Tuendeleye Pamoja project, and the Integrated Project for Agricultural Growth in the Great Lakes (PICAGL) in South Kivu Province.

## 3. Materials and Methods

### 3.1. Description of the Study Area

This study was conducted in South Kivu Province in the eastern Democratic Republic of Congo (Figure 2). A large majority (70%) of the inhabitants of South Kivu work in agriculture, including crop farming, fishing, and livestock rearing. About 80% of the areas in South Kivu have the potential for cassava production, and cassava is the most consumed food in both rural and urban areas [25,31,44]. The province has climatic conditions favorable to cassava and other crops, as average temperatures range from 11 °C to 25 °C [44]. It is also open to other provinces and countries. To the south, it borders Tanganyika Province, while to the north, it borders North Kivu Province. To the east, it borders the Republic of Rwanda, Burundi, and Tanzania, while to the west, it borders Maniema Province. Its 5.8 million people (2015 estimate) represent an important demand for local production.

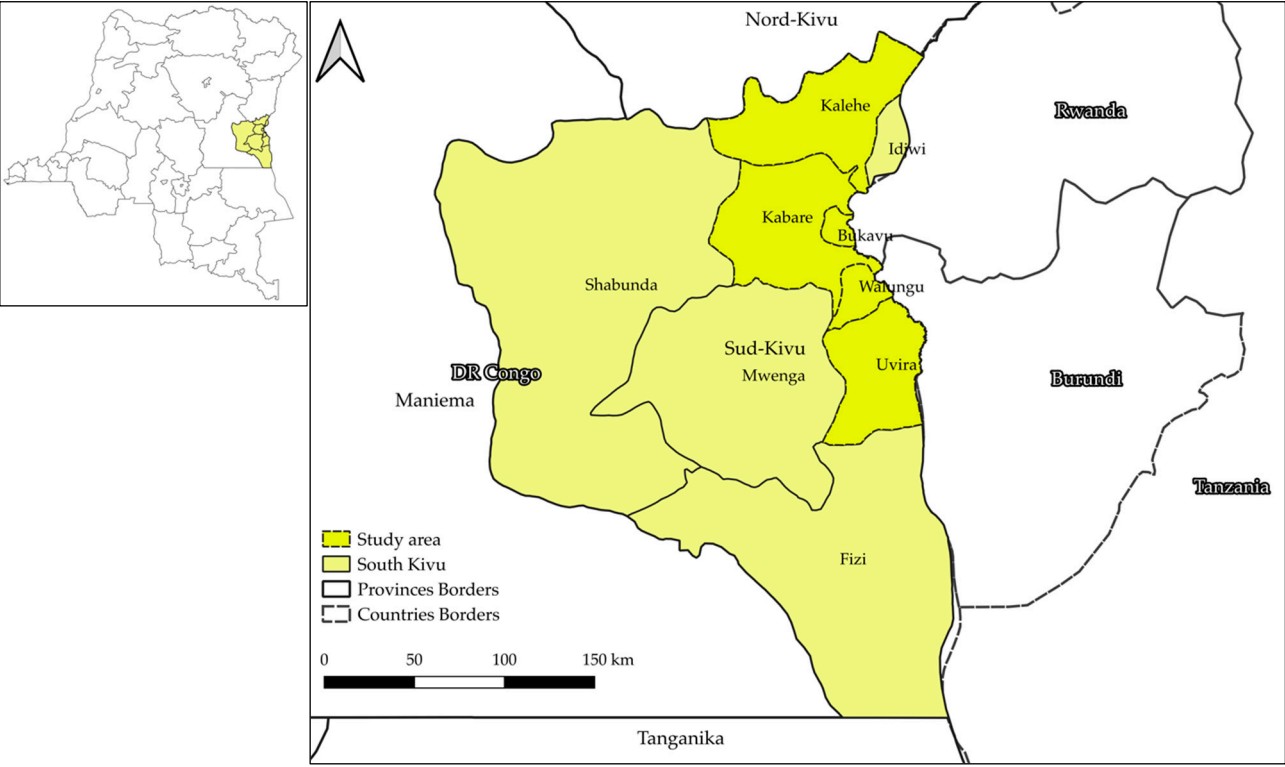

**Figure 2.** Map of South Kivu Province, the study area. Source. Authors' conception using QGIS 3.10.

### 3.2. Data Description and Sampling Procedure

The data used in this paper were collected in two surveys. First, we surveyed youth graduates in September 2019 to identify nonagricultural jobs that are comparable to CCPCs and attractive to recent university graduates. Second, we surveyed agribusiness and non-agribusiness companies in November 2019, considering CCPCs as a case study of agribusiness.

We used a multistage sampling approach for both surveys. In the first survey, three local universities in Bukavu were purposively selected based on their size as measured by the number of students in the first stage. In the second stage, approximately 30 youth graduates were randomly selected at each university, with half male and half female respondents (see Table 1 for a description of the graduates surveyed). The reason for this survey was that young graduates have a certain preference for the labor market and are mainly affected by unemployment, so they can express their preferences for comparable jobs outside the agricultural sector. After compiling the data, five comparable occupations outside the agricultural sector were selected, including stores and magazines preferred by 70% of young graduates, hairdressers preferred by 44%, restaurants and bistros preferred by 43%, pharmacies preferred by 39%, and sewing workshops preferred by 25% of young graduates, based on their relative preference and were compared with CCPCs (Table 1).

**Table 1.** Youth graduate characteristics and preferred non-agribusiness enterprises.

|  | UCB | UEA | UOB | Overall |
|---|---|---|---|---|
| Sample size | 32 | 30 | 31 | 93 |
| Sex (male = 1) | 0.44 | 0.57 | 0.48 | 0.49 |
| Age (years) | 23.44 | 24.23 | 23.68 | 23.77 |
| Marital status (married = 1) | 1.06 | 1.07 | 1.03 | 1.05 |
| Household size (Persons) | 7.72 | 7.27 | 6.94 | 7.31 |
| Religion (non-Christian = 1) | 0.09 | 0.03 | 0.06 | 0.06 |
| Field study |  |  |  |  |
|   Agronomy (yes = 1) | 0.38 | 0.37 | 0.00 | 0.25 |
|   Economics (yes = 1) | 0.25 | 0.33 | 0.13 | 0.24 |
|   Medicine (yes = 1) | 0.13 | 0.13 | 0.74 | 0.33 |
|   Social science (yes = 1) | 0.00 | 0.17 | 0.06 | 0.08 |
|   Others (yes = 1) | 0.25 | 0.00 | 0.06 | 0.11 |
| Preferred non-agribusiness activities |  |  |  |  |
|   Shop and magazine (yes = 1) | 0.72 | 0.67 | 0.71 | 0.70 |
|   Pharmacies (yes = 1) | 0.16 | 0.23 | 0.77 | 0.39 |
|   Mechanic (yes = 1) | 0.06 | 0.27 | 0.10 | 0.14 |
|   Sewing workshops (yes = 1) | 0.16 | 0.40 | 0.23 | 0.26 |
|   Restaurant and bistro (yes = 1) | 0.22 | 0.37 | 0.71 | 0.43 |
|   Hairdressers (yes = 1) | 0.19 | 0.63 | 0.52 | 0.44 |
|   Transport (yes = 1) | 0.16 | 0.33 | 0.16 | 0.22 |

Source: Authors' analysis. Notes. Ref. = reference. Other fields of study include law studies and informatics; they have been aggregated because of an insignificant number of surveyed students. UCB = Université Catholique de Bukavu, UEA = Université Évangélique en Afrique, UOB = Université Officielle de Bukavu.

In the second survey, territories where the CCPC model was implemented by the IITA since 2013 in South Kivu, were purposively selected, at the first stage, and all of the seven operational CCPCs at the time of data collection were sampled in the four territories, including Walungu (two CCPCs), Kabare (two CCPCs), Kalehe (one CCPC), and Uvira (two CCPCs), at the second stage. Considering non-agribusiness, a total of 30 non-agribusiness enterprises were randomly selected for each non-agribusiness pre-identified enterprise in Bukavu, at the first level, giving a total of 150 non-agribusiness enterprises. The manager was the target respondent for CCPCs or otherwise the main technician, and it was the enterprise owner for non-agribusiness activities or otherwise the manager. Both surveys were implemented by trained enumerators who had conducted similar surveys in the past, spoke the local language, and had a minimum of a BSc. level. Tablets with a pre-loaded electronic questionnaire were used. We combined both face-to-face interviews for accessible areas of data collection and phone call interviews for non-accessible areas, including Kalehe and Uvira territories, due to a large distance and transport constraints. One supervisor was trained to check the quality of completed questionnaires and make corrections, if required, daily at the field level. Besides, the data were assembled at the ona central server (https://ona.io), where a second checking of the data quality was conducted instantly, and possible mistakes were brought back to the attention of the supervisor for correction in the field before the enumerators moved from one area to the next. This paper is based on the analysis of the cross-sectional data collected at the firm-level.

### 3.3. Empirical Approach

This study employed technical efficiency and profitability as competitiveness proxies to compare agribusiness and non-agribusiness competitivity, following [32,33,45–49]. Regarding the technical efficiency analysis, we implemented a Data Envelopment Analysis (DEA), while the profitability analysis was based on two indicators, including the cost–benefit ratio and the net monthly revenue.

### 3.3.1. Technical Efficiency Analysis Approach

The DEA is a popular tool used to measure the efficiency of decision-making units (DMUs) that have multiple inputs and outputs [45,46,48,50,51]. Conceptually, efficiency refers to the output that a company can produce given the available inputs or a specific combination of inputs. DMUs set may be manufacturers, retailers, and service companies. In our case, DMUs refer to CCPCs and non-agribusiness enterprises. Specifically, the choice of this method is motivated by the fact that econometric models with a small number of observations, as was the case in this study, may turn out to be inefficient and unstable. Furthermore, when it is used to compare sectors or activities, entrepreneurs can learn from other activities, which is more important in the case of agribusiness and non-agribusiness [45,52–55].

Two models of DAE are usually used in the literature. The first model, the CCR (Charnes, Cooper, and Rhodes) model, assumes constant returns to scale (CRS) so that a change in the input level leads to an equiproportionate change in the output level [56]. The second model is the BCC (Banker, Charnes, and Cooper) model, which assumes that variables return to scale (VRS) where performance is bounded by a piecewise linear frontier [57]. The CCR model identifies overall technical efficiency (pure technical efficiency and scale efficiencies), while the BCC model identifies the pure technical efficiency only [45]. These two specifications were used in this study.

The DEA-CCR uses the following theoretical model for evaluating DMU's efficiency:

$$\text{Maximize } h_{jo} = \frac{\sum_{r=1}^{s} u_r y_{rjo}}{\sum_{i=1}^{m} v_i x_{ijo}} \tag{1}$$

$$\text{Subject to } \frac{\sum_{r=1}^{s} u_r y_{r_j}}{\sum_{i=1}^{m} v_i x_{ij}} \leq 1, \qquad j = 1, \ldots, n$$

$$\mu_i, \, v_r \geq 0, \qquad i = 1, \ldots, m \qquad r = 1, \ldots, s$$

where $h_{jo}$ is the DEA technical efficient score; $y_{rj}$ is the amount of output $r$ from DMU $j$; $x_{ij}$ is the amount of input $i$ from unit $j$; $\mu_r$ and $v_i$ are the weight of outputs and inputs, respectively; $n$ represents the total number of DMU; $s$ is the total number of outputs; $m$ is the total number of inputs. This mathematical formulation of DEA considers $n$ DMUs, when each DMU $j$ ($j = 1, \ldots, n$) uses $m$ inputs $x_j = \left( x_{1j}, \cdots, x_{m_j} \right) > 0$ for producing $s$ outputs $y_j = \left( y_{1j}, \cdots, y_{m_j} \right) > 0$.

The weights are all positive, and ratios are bounded by 100%. If a DMU reaches the max possible value of 100%, it is considered fully efficient for $h_{jo} = 1$, or it is less efficient or inefficient for $h_{jo} = 0$. Equation (1) can be translated into a linear program (Equation (2)), which can be solved relatively easily, and a DEA solves $n$ linear programs, one for each unit:

$$\text{Maximize } h_{jo} = \sum_{r=1}^{s} u_r y_{r_{jo}} \tag{2}$$

$$\text{Subject to : } \sum_{i=1}^{m} v_i x_{ijo} = 1,$$

$$\sum_{r=1}^{s} u_r y_{r_j} - \sum_{i=1}^{m} v_i x_{ij} \leq 0, j = 1, \ldots, n$$

$$\mu_i, \, v_r \geq \varepsilon \text{ for } i = 1, \ldots, m \text{ and } r = 1, \ldots, s$$

where $\varepsilon \geq 0$ is a non-Archimedean number that is smaller than any non-negative real number. Other parameters are specified as in Equation (1).

The DAE-BCC model can be defined by adding the constraint $z_{jo}$ in Equation (2), as shown in Equation (3).

$$\text{Maximize } h_{jo} = \sum_{r=1}^{s} u_r y_{r_{jo}} + z_{jo} \tag{3}$$

$$\text{Subject to}: \sum_{i=1}^{m} v_i x_{ijo} + z_{jo}$$

$$\sum_{r=1}^{s} u_r y_{r_j} - \sum_{i=1}^{m} v_i x_{ij} + z_{jo} \leq 0, j = 1, \ldots, n$$

$$\mu_i, v_r \geq \varepsilon \; for \; i = 1, \ldots, m \; and \; r = 1, \ldots, s$$

The number of decision-making units (DMUs) increases with the number of degrees of freedom and decreases with the number of inputs and outputs. The authors of [58] proposed a rule to count the DMUs number.

$$\xi \geq \max\{\gamma x \Phi, 3(\gamma + \Phi)\} \tag{4}$$

where $\xi$ = DMUs, $\gamma$ = inputs, and $\Phi$ = outputs.

The number of inputs $x_{ij}$ included labor and capital factors, following [45,54,59]. The number of labors was measured in terms of the number of persons that regularly work in the enterprise and are remunerated by the enterprise. We, therefore, excluded the unpaid labor force, such as family labor and help from a friend, for which we could not estimate the cost. Fixed assets were measured in terms of physical capital, such as peeling or cutting machines for CCPCs, and other assets, such as telephones, tablet computing, computers, calculators, fridges, televisions, or any asset that concurs in the production process of non-agribusiness enterprises.

The amount of output $y_{rj}$ included turnover, the quantity of cassava root processed for CCPCs, or the number of articles averagely sold for non-agribusiness activities. Similar outputs were used by [48,60–62]. The turnover was an approximative amount of money that the enterprise generated monthly before the survey and is expressed in US dollars (US \$1 = 1800 CDF). The quantity of cassava root that the CCPC averagely processes under normal circumstances, such as availability of cassava root, electricity, and all machines being operational, was expressed in kilograms. The number of articles averagely sold was considered as the output for the non-agribusiness enterprise. Stata software was used to derive solutions for Equations (3) and (4).

### 3.3.2. Profitability Analysis Approach of Agribusiness and Non-Agribusiness

To determine the profitability of agribusiness and non-agribusiness activities, the cost–benefit ratio was carried out, following [32,33,63]. The cost–benefit ratio is the ratio between turnover and cost, measured monthly. The assumption underlying this profitability metric is that the higher the cost–benefit ratio, the higher the profit made by the enterprise. Besides, given the cost and turnover, we estimated the net revenue by differentiating the total monthly turnover and the total monthly cost. The total cost was captured as a percentage of the turnover. This approach made feasible the estimation of the cost that the enterprise supports during a given period of activity.

The mathematical formulation for the analysis of an activity $a$ is presented below:

$$T = \sum_{a=1}^{n} q_a p_a \tag{5}$$

$$C = \% * \left( \sum_{a=1}^{n} q_a p_a \right) \tag{6}$$

$$B/C = \frac{\left[ \sum_{a=1}^{n} q_a p_a - \left( \% * \left( \sum_{a=1}^{n} q_a p_a \right) \right) \right]}{\% * \left( \sum_{a=1}^{n} q_a p_a \right)} \tag{7}$$

$$nr = T - C \tag{8}$$

where B/C represents the benefit/cost ratio, which represents the amount of money an investor receives from each dollar invested in an activity; T is the monthly turnover; C is the total monthly cost; $p_a$ is the price of a unit sold; $q_a$ represents the quantity sold; nr denotes the net monthly revenue. All these parameters were captured considering one month.

### 3.3.3. Econometrical Approach for Estimating Determinants of Technical Efficiency

We acknowledge that managers' and other activity's specific characteristics could determine activity technical efficiency beyond inputs that were considered under the DEA-CCR and DAE-BCC. Besides, the DEA does not allow one to judge the significance of the difference between agribusiness and non-agribusiness technical efficiency. To overcome these limits, we performed a linear regression and considered, as an outcome of interest, the activities $i$'s overall technical efficiency, noted as $y_{rj}$ Formally, we estimated the following models:

$$y_{rj} = \zeta_i + \psi A_i + \varepsilon_i \tag{9}$$

$$y_{rj} = \zeta_i + \psi A_i + \theta V_m + \omega Z_i + \varepsilon_i \tag{10}$$

where activity $A_i$ takes value 1 if it is an agribusiness and 0 if it is otherwise. $V_m$ is a vector of manager $m$ characteristics, including age, sex, marital status, education, experience, and professional training. $Z_i$ is a vector of the enterprise's $i$ characteristics, such as number of machines, operational cost, number of workers, usage of ICT (Information and Communication Technologies), access to water, access to electricity, access to finance, and number of paid taxes; $\zeta$, $\psi$ and $\omega$ are parameters to be estimated; and $\varepsilon$ is the error term.

Equation (9) allowed us to determine the technical efficiency differential $\psi$ between an agribusiness and non-agribusiness. This equation was considered as our main econometrical estimation, as it is an off-endogenous problem that could normally occur in Equation (10), as both vectors $V$ and $Z$ are correlated with activity. We estimated Equations (9) and (10) using the ordinary least square. Furthermore, we used a stepwise approach that allowed us to examine the significance of each vector using various specifications of Equation (10) [64–66].

### 3.3.4. Variables Description

*Activity*: we used an aggregated variable that takes value 1 if the activity is an agribusiness (CCPC) and 0 if it is otherwise to capture the technical efficiency variation between agribusiness and non-agricultural activities. Based on the fact that the majority of youths have stayed out of agribusiness, one could hypothesize that agribusiness lacks efficiency and, hence, is less attractive to them compared to other job opportunities [15,32–34].

*Age of the manager*: we also considered the age of the manager, or the owner, of activities on the technical efficiency matter. The latter is somewhat associated with the decision to engage in agribusiness, on the one side, and experience, on the other side. Besides, agribusiness (CCPC) requires more physical effort compared to non-agricultural activities considered in this study; therefore, we may expect that agribusiness's technical efficiency increases for male managers. Moreover, we considered the marital status of the manager, as one could hypothesize that married workers tend to be more engaged and hence more efficient [60–62,67].

*Education of the manager:* as a component of human capital, education plays a significant role in labor productivity. In fact, the more managers are educated, the more performant they are. We considered also professional training as an additional source of knowledge and as a human capital subcomponent that increases efficiency on a task. Hence, both education and professional training are expected to be positively associated with technical efficiency [48,59,66].

*Access to credit:* a lack of access to funds has been emphasized from time to time as the main constraint for youth engagement in agribusiness, but access to finance also plays a crucial role during activities' growing process [9]. In fact, access to credit may be correlated to technical efficiency through equipment acquisition or input prepayment, such as cassava root, which could guarantee their availability regularly. Thus, we hypothesized that access to credit would be positively correlated to technical efficiency.

Other variables were included in the model, such as ICT usage measured in terms of the usage of technologies of information and communication (TICs) in activities operations, and they included telephones, computers, or local medias (radios). TICs are the channel

through which managers can be connected to clients or input suppliers. They can be seen also as channels through which knowledge and other information, such as price information and regulations, can be obtained and hence shape the behavior of managers. Access to water and electricity is important for activities such as hairdressers or CCPCs as they enter into the production process. Electricity is the efficient source of energy for many activities, including CCPCs, while water intervenes mainly during cassava root preparation, and its availability could be associated with agribusiness technical efficiency. Finally, the number of paid taxes was measured in terms of the number taxes that the activity paid during the last 12 months preceding the survey [15].

## 4. Results and Discussion

### 4.1. Descriptive Statistics of the Manager and Enterprise's Characteristics

The descriptive statistics of respondents are summarized in Table 2. It shows that all respondents in both agribusiness and non-agribusiness enterprises were aged less than 45 years old. They were dominantly constituted by males, as 71% and 72% of respondents were males in agribusiness and non-agribusiness, respectively. The graduate education level (80% of respondents) was more represented in agribusiness activities, while the secondary education level appeared the most representative in non-agribusiness (45% of respondents). This figure highlights that agribusiness is not always for non-educated people from rural areas [15]. Married respondents were more predominant in agribusiness activities than non-agribusiness activities, with 86% and 39% of the respondents, respectively. Around nine out of 10 enterprises were using ICT tools, and less than two enterprises out of 10 accessed credit services in agribusiness and non-agribusiness activities. The majority of CCPCs had no access to electricity (71%) while more than 70% of non-agribusiness activities had access to electricity. On the contrary, many agribusiness managers had access to professional training (71%) compared to non-agribusiness managers. The chi-square highlights areas of significant differences between agribusiness and non-agribusiness enterprises.

**Table 2.** Descriptive statistics of respondents and enterprise's characteristics.

| Variables | | Activities | | Overall | Chi-2 |
|---|---|---|---|---|---|
| | | Agribusiness (%) | Non-Agribusiness (%) | | |
| Age of the manager | 18–24 | 0.0 | 23.0 | 21.8 | 23.288 *** |
| | 25–34 | 0.0 | 57.0 | 54.2 | |
| | 35–45 | 100.0 | 20.0 | 23.9 | |
| Sex of the manager | Male | 71.4 | 71.9 | 71.8 | 0.001 |
| | Female | 28.6 | 28.2 | 28.2 | |
| Manager Education level | Primary | 0.0 | 6.8 | 6.5 | 4.446 *** |
| | Secondary | 20.0 | 45.1 | 44.2 | |
| | Graduate | 80.0 | 34.6 | 36.2 | |
| | License | 0.0 | 13.5 | 13.0 | |
| Manager Marital status | Non-married | 14.3 | 60.7 | 58.5 | 5.914 *** |
| | Married | 85.7 | 39.3 | 41.6 | |
| ICT usage | Yes | 85.7 | 89.6 | 89.4 | 0.103 |
| | No | 14.3 | 10.5 | 10.6 | |
| Access to credit | Yes | 14.3 | 15.6 | 15.5 | 0.008 |
| | No | 85.7 | 84.4 | 84.5 | |
| Access to electricity | Yes | 28.3 | 74.8 | 72.5 | 7.144 *** |
| | No | 71.4 | 25.2 | 27.5 | |
| Professional Training | Yes | 71.4 | 27.4 | 29.6 | 6.192 ** |
| | No | 28.6 | 72.6 | 70.4 | |

Source: Authors' analysis. Notes. ***, and ** are coefficients significant at 1%, and 5%, respectively.

### 4.2. Overall Technical Efficiency of CCPCs and Non-Agribusiness Enterprises

Considering the technical efficiency of different activities, Table 3 provides the result from the DEA-CCR and DEA-BCC models, respectively, for agribusiness and non-agribusiness. The most important findings in Table 3 are as follows.

**Table 3.** The overall efficiency of CCPCs and non-agribusiness enterprises.

| Technical Efficiency | Agribusiness | | Non-Agribusiness Activities | |
| --- | --- | --- | --- | --- |
| | **DEA-BCC** | **DEA-CCR** | **DEA-BCC** | **DEA-CCR** |
| | **Freq. (%)** | **Freq. (%)** | **Freq. (%)** | **Freq. (%)** |
| <0.5001 | | | 72 (56.69) | 81 (63.28) |
| 0.5001–0.6000 | 2 (28.57) | 2 (28.57) | 5( 3.94) | 8 (6.25) |
| 0.6001–0.7000 | | | 8 (6.30) | 5 (3.91) |
| 0.7001–0.8000 | | 2 (28.57) | 1 (0.78) | 4 (3.13) |
| 0.8001–0.9000 | 1 (14.29) | | 7 (5.51) | 5 (3.91) |
| 0.9001–0.9999 | | | 2 (1.57) | 1 (0.78) |
| 1 | 4 (57.14) | 3 (42.86) | 32 (25.20) | 24 (18.75) |
| Total | 7 (100) | 7(100) | 127 (100) | 128 |
| Minimum | 0.56 | 0.52 | 0.02 | 0.19 |
| Maximum | 1 | 1 | 1 | 1 |
| Mean | 0.86 | 0.80 | 0.51 | 0.44 |
| Standard Deviation | 0.20 | 0.20 | 0.36 | 0.34 |

Source: Authors' analysis. Notes. CCPCs are identified by the names of their location.

First, the DEA-BCC yielded a higher average efficiency estimate than the DEA-CCR model, as one could expect. The reason is that the DEA model, with an assumption of constant returns to scale, estimates the purely technical and scale efficiency taken together, while a DEA model, with the assumption of variable returns to scale, identifies technical efficiency alone, as found by [45,46].

Secondly, four and three agribusiness CCPCs out of seven were fully efficient under BCC and CCR, as their overall technical efficiency was equal to 1. Hence, when comparing the number of workers and the fixed assets with the quantity of cassava root processed and turnover, four and three CCPCs were efficient under the DEA-BCC and DEA-CCR models, respectively. In terms of percentage, more than 40% of CCPCs were efficient, as their efficient score equaled 1. Secondly, and considering non-agribusiness enterprises, only 32 and 24 non-agribusiness enterprises out of 127 and 128 non-agribusiness enterprises were fully efficient under the DEA-BCC and DEA-CCR models, respectively. They represent 25% and 19% of non-agribusiness activities, meaning that when comparing the number of workers and the fixed assets with the number of articles sold and the turnover, less than 30% of non-agribusiness enterprises are efficient.

Fourth, the average overall technical efficiency when DEA-BCC and DEA-CCR were applied was 86% and 80% for agribusiness CCPCs, while it was 51% and 44% for non-agribusiness enterprises, meaning that, on average, the CCPCs analyzed could operate at 86% and 80% of their current levels while still returning the same output value, while non-agribusiness enterprises could operate at 51% and 44%. Finally, these findings demonstrate that substantial inefficiency occurred in the non-agribusiness enterprises, implying that some of the non-agribusiness managers were not operating at an efficient scale, and improvement in the overall efficiencies could be achieved if the managers adjusted their scale of operations. Moreover, agribusiness remains efficient compared to non-agribusiness under the DEA-BCC and DEA-CCR models.

These results can be explained by the fact that agribusiness managers are aged and have a higher human capital, captured in terms of education level and professional training, compared to non-agribusiness managers, on average. These explanations are consistent with the findings of [48,59,66], which showed that knowledge and experience related to age are relevant factors in decision-making about inputs selection, allocation, and utilization,

on the one hand, and workers' productivity increases with human capital, on the other. Moreover, we could hypothesize that being married can lead to good managerial behavior, as agribusiness managers were found to be married compared to non-agribusiness managers.

Focusing on the specific activities, Table 4 shows that Mulamba, Kamanyola, and Kavumu CCPCs were fully efficient CCPCs when both the DEA-BCC and DEA-CCR models were performed, while the nonspecific non-agribusiness enterprise was fully efficient. This result can be explained by the geographical (located in a plain area and sharing the border with Rwanda and Burundi) and climatical conditions of the location of these CCPCs. In their areas, more than 50% of farmers grow cassava, which constitutes a regular source of cassava roots.

**Table 4.** The overall efficiency of specific CCPCs and non-agribusiness enterprises.

| Technical Efficiency | | Cassava Community Processing Centers | | | | | | |
|---|---|---|---|---|---|---|---|---|
| | | Mulamba | Katana | Sange | Luvungi | Kichanga | Kamanyola | Kavumu |
| DEA-BCC | <0.5001 | | | | | | | |
| | 0.5001–0.6000 | | | X | X | | | |
| | 0.6001–0.7000 | | | | | | | |
| | 0.7001–0.8000 | | | | | | | |
| | 0.8001–0.9000 | | X | | | | | |
| | 0.9001–0.9999 | | | | | | | |
| | 1 | X | | | | X | X | X |
| DEA-CCR | <0.5001 | | | | | | | |
| | 0.5001–0.6000 | | | X | X | | | |
| | 0.6001–0.7000 | | | | | | | |
| | 0.7001–0.8000 | | X | | | X | | |
| | 0.8001–0.9000 | | | | | | | |
| | 0.9001–0.9999 | | | | | | | |
| | 1 | X | | | | | X | X |

| Technical Efficiency | | Non-Agribusiness | | | | |
|---|---|---|---|---|---|---|
| | | Hairdresser | Pharmacies | Sewing Workshop | Restaurant and Bistro | Shop and Magazine |
| DEA-BCC | <0.5001 | X | | X | | |
| | 0.5001–0.6000 | | X | | | |
| | 0.6001–0.7000 | | | | | X |
| | 0.7001–0.8000 | | | | X | |
| | 0.8001–0.9000 | | | | | |
| | 0.9001–0.9999 | | | | | |
| | 1 | | | | | |
| DEA-CCR | <0.5001 | X | X | X | | |
| | 0.5001–0.6000 | | | | | X |
| | 0.6001–0.7000 | | | | X | |
| | 0.7001–0.8000 | | | | | |
| | 0.8001–0.9000 | | | | | |
| | 0.9001–0.9999 | | | | | |
| | 1 | | | | | |

Source: Authors' analysis. Notes. CCPCs are identified by the names of their location.

### 4.3. Determinants of the Overall Technical Efficiency

Table 5 presents the drivers of the overall technical efficiency under DEA-BCC, with variable returns to scale (VRS), and DEA-CCR, with a constant return to scale (CRS). Columns (1) and (5) are estimation results of Equation (9), while other columns are the various estimations of Equation (10). The DEA-BCC and DEA-CCR models indicate that, on average, the efficiency level in agribusiness is higher compared to non-agribusiness (NA) activities. The technical efficiencies of agribusiness (CCPC) were 0.321 (BCC) and 0.317 (CCR) points higher compared to non-agricultural businesses. The differences were high and re-

mained stable when we controlled for activity and manager characteristics, reaching 0.791 and 0.737, respectively, when DEA-BCC and DEA-CCR were performed (Columns 4 and 8). These econometrical estimations are consistent with the above-mentioned descriptive results.

**Table 5.** The drivers of the Overall technical efficiency under the DEA-BCC and DEA-CCR.

| Variables | Dependent Variable: Overall Technical Efficiency | | | | | | | |
| --- | --- | --- | --- | --- | --- | --- | --- | --- |
| | DEA-BCC | | | | DEA-CCR | | | |
| | (1) | (2) | (3) | (4) | (5) | (6) | (7) | (8) |
| Activity (Agribusiness) | 0.321 ** | 0.493 *** | 0.555 *** | 0.791 *** | 0.317 ** | 0.487 *** | 0.518 ** | 0.737 *** |
| | (0.141) | (0.158) | (0.206) | (0.215) | (0.136) | (0.153) | (0.201) | (0.210) |
| Age (Years) | | −0.003 | | 0.006 | | −0.004 | | 0.004 |
| | | (0.006) | | (0.006) | | (0.006) | | (0.006) |
| Sex (Male) | | 0.123 * | | 0.108 * | | 0.126 * | | 0.115 * |
| | | (0.068) | | (0.063) | | (0.066) | | (0.061) |
| Education level | | | | | | | | |
| Primary | | 0.277 | | 0.186 | | 0.311 | | 0.213 |
| | | (0.224) | | (0.210) | | (0.217) | | (0.205) |
| Secondary | | 0.365 * | | 0.220 | | 0.384 ** | | 0.242 |
| | | (0.196) | | (0.184) | | (0.189) | | (0.180) |
| Bachelor | | 0.370 * | | 0.233 | | 0.381 ** | | 0.248 |
| | | (0.192) | | (0.180) | | (0.186) | | (0.176) |
| Master | | 0.370 * | | 0.221 | | 0.363 * | | 0.206 |
| | | (0.208) | | (0.196) | | (0.201) | | (0.192) |
| Marital status (Married) | | 0.054 | | −0.066 | | 0.074 | | −0.029 |
| | | (0.082) | | (0.081) | | (0.080) | | (0.080) |
| Experience (Years) | | −0.027 * | | −0.022 | | −0.027 * | | −0.022 |
| | | (0.016) | | (0.016) | | (0.016) | | (0.015) |
| Experience square (Years) | | 0.001 | | 0.000 | | 0.001 | | 0.000 |
| | | (0.001) | | (0.001) | | (0.001) | | (0.001) |
| Professional training (Yes) | | 0.135 ** | | 0.183 *** | | 0.125 * | | 0.175 *** |
| | | (0.067) | | (0.066) | | (0.065) | | (0.064) |
| Number of machines | | | −0.023 * | −0.018 | | | −0.024 ** | −0.019 |
| | | | (0.012) | (0.013) | | | (0.012) | (0.012) |
| Number of workers | | | 0.004 | 0.003 | | | 0.005 | 0.005 |
| | | | (0.009) | (0.009) | | | (0.009) | (0.009) |
| Operational cost | | | −0.000 | −0.000 * | | | −0.000 | −0.000 * |
| | | | (0.000) | (0.000) | | | (0.000) | (0.000) |
| Use of ICT (Yes) | | | −0.016 | −0.130 | | | −0.031 | −0.137 |
| | | | (0.091) | (0.096) | | | (0.089) | (0.094) |
| Access to finance (Yes) | | | −0.093 | −0.075 | | | −0.076 | −0.051 |
| | | | (0.083) | (0.082) | | | (0.081) | (0.080) |
| Access to water (Yes) | | | 0.090 | 0.118 * | | | 0.089 | 0.113 * |
| | | | (0.067) | (0.065) | | | (0.065) | (0.063) |
| Access to electricity (Yes) | | | 0.282 *** | 0.286 *** | | | 0.255 *** | 0.249 *** |
| | | | (0.068) | (0.067) | | | (0.066) | (0.066) |
| Number of paid taxes | | | 0.031 ** | 0.033 ** | | | 0.032 ** | 0.035 ** |
| | | | (0.015) | (0.014) | | | (0.014) | (0.014) |
| Constant | 0.538 *** | −0.002 | 0.266 ** | −0.255 | 0.516 *** | −0.032 | 0.269 ** | −0.261 |
| | (0.031) | (0.284) | (0.115) | (0.279) | (0.030) | (0.275) | (0.112) | (0.273) |
| Observations | 142 | 142 | 142 | 142 | 142 | 142 | 142 | 142 |
| R-squared Adjusted | 0.029 | 0.113 | 0.159 | 0.267 | 0.030 | 0.114 | 0.150 | 0.252 |

Source: Authors' analysis. Notes. ***, **, and * are coefficients significant at 1%, 5%, and 10%, respectively; figures in parenthesis are normal standard errors. No education was considered as the reference modality.

Let us emphasize that we considered, in this study, Cassava, which is the main agricultural product in the region. Overall, the results imply that investing one dollar in this agribusiness is technically efficient as compared to non-agricultural activities. We assume that these results might be generalized, since, in the spirit of the comparative advantage theory, it is always efficient for a region to specialize in a product that is produced with

lower relative costs (and that yields a higher revenue). In fact, the climatic conditions of the region are favorable to produce Cassava, and there is a permanent demand for the product, since it is among the main food consumed by the community. Nonetheless, the Cassava Community Processing Centers require important fixed costs, which might constitute an entry barrier for young entrepreneurs. Given the strategic position of Cassava in the community, public and private funding should be mobilized to attract young people to this amazing agribusiness.

Considering the manager's characteristics, Table 5 shows that the overall technical efficiency increased significantly by 0.123 and 0.126 (Columns 2 and 6) when the manager was a male compared to a female. Moreover, professional training had a significant influence on overall technical efficiency. Being trained increased the overall technical efficiency by 0.135 and 0.183, respectively, in Columns (2) and (3) and by 0.125 and 0.175 in Columns (6) and (8). Besides, the manager being educated above primary school had a positive effect on the efficiency (Columns 2 and 6). These results are consistent with the findings of several previous studies, which found that education enhances technical efficiency [48,59,66].

The technical efficiency decreased by 0.023 and 0.024 for additional fixed assets, respectively, in Columns (2) and (6), supporting the argument that agribusiness activity is less intensive in physical capital but intensive in human capital. The latter has been evidenced as a relevant determinant of efficiency by [60–62,67]. Finally, other characteristics, such as the number of paid taxes, access to water, and electricity, significantly increased the overall efficiency. This magnitude of the increase in the technical efficiency was even more pronounced for access to electricity, with 0.282 and 0.286 in Columns (3) and (4) and 0.255 and 0.249 in Columns (7) and (8), demonstrating the relevant side of the access to electricity, especially for CCPCs. Other factors had a marginal effect.

### 4.4. Profitability of Agribusiness and Non-Agribusiness Activities

As can be seen from Table 6, the agribusiness net monthly revenue was 814 US dollars against 104.1 US dollars in non-agribusiness. The undiscounted benefit/cost ratios were 2.8 and 2.1 for agribusiness and non-agribusiness, respectively. These results show that agribusiness generates high earns compared to non-agribusiness. Furthermore, every dollar invested provides 2.8 US dollars to the agribusiness's investor, while non-agribusiness provides 2.1 US dollars in return. Among agribusiness CCPCs, Kavumu CCPC presents the highest level of monthly net revenue, with 2400 US dollars, while Kamanyola CCPC provides a higher return (5.0). Among non-agribusiness activities, shops and magazines are on top of the list, considering the two profitability indicators (206 US dollars of net monthly revenue and 5.0 benefit/cost ratio). These findings come to complete the existent literature on the socio-economic benefits of cassava and related derivates products, as well as other agribusiness cases [32,33,42,43,63].

**Table 6.** Profitability analysis of Agribusiness and Non-agribusiness activities.

| Activities | Net Revenue (US $) | Benefit/Cost Ratio |
|---|---|---|
| Agribusiness | 814 | 2.8 |
| Mulamba CCPC | 108 | 1.8 |
| Katana CCPC | 168 | 3.3 |
| Sange CCPC | 360 | 1.4 |
| Luvungi CCPC | 480 | 1.4 |
| Kichanga CCPC | 900 | 4.0 |
| Kamanyola CCPC | 1280 | 5.0 |
| Kavumu CCPC | 2400 | 2.5 |
| Non-agribusiness activities | 104.1 | 2.1 |
| Hairdresser | 49 | 1.8 |
| Pharmacies | 62 | 1.9 |
| Sewing workshop | 75 | 1.7 |
| Restaurant and Bistro | 136 | 2.2 |
| Shop and magazine | 206 | 2.6 |

Source: Authors' analysis. Notes. CCPCs are identified by the names of their location.

Concretely, these results show that youths entering agribusiness, and especially Cassava Processing, may earn more compared to non-agricultural businesses. Moreover, they provide answers on what works on the ground [22,32–34], on the one hand, and constitute a convincing argument for agribusiness promotion as a solution to youth unemployment, on the other.

## 5. Conclusions and Recommendations

The study examined the competitiveness of cassava community processing centers compared to non-agribusiness attractive activities using technical efficiency and profitability as proxies for competitiveness in South Kivu, eastern DRC. The results revealed that cassava business is statistically higher in efficiency than non-agribusiness activities. Among cassava community processing centers, the fully efficient CCPCs were Mulamba, Kavumu, and Kamanyola. Furthermore, cassava community processing centers were found to be more profitable than non-agribusiness enterprises. Kamanyola CCPC was the most profitable CCPC among all of the CCPCs in the study area.

Regarding factors that determine technical efficiency, the regression results showed that the overall technical efficiency increased for agribusiness and remained stable when controlling for activities and managers' characteristics. Besides, the estimated models show that the overall technical efficiency increased with education, professional training, access to electricity, access to water, and the sex of the manager. Finally, profitability analysis demonstrated that agribusiness yields, on average, a higher net revenue compared to non-agribusiness enterprises and provides, consequently, a higher return on investment.

These findings demonstrate that agribusiness can be the best choice for youths and remains a suitable solution to youth unemployment in the region. However, based on these findings, the following recommendations can be formulated for the sustainability of this agribusiness model. First, efforts should be geared towards the capacity building of CCPCs managers in areas like stock and supply management, innovation, and value addition through the processing of cassava in various derivates products and marketing. CCPC owners should devote more resources to access to electricity and water, if it is available in the village or territory, or local authorities should improve access to electricity and water for a higher attractiveness of agribusiness. The delivery of available cassava technologies on the ground should allow for more availability of cassava roots. Finally, the formalization of activities, such as having an RCCM number ("Registre du Commerce et du Credit Mobilier"), should be perceived as an incentive for youth entrepreneurs to do well.

We should encourage youths to engage in cassava processing units, since it is more profitable than comparable non-agribusiness activities. With these changes, and with

appropriate food system-related policy, investment, and innovation, new economic and employment opportunities can emerge for youths. This study has some limitations. First, by not considering many agribusiness cases in the analysis, the conclusions presented here are limited to a specific case of agribusiness, cassava processing units, and not all. Besides, non-agribusiness activities are mostly self-employment and are subjectively selected by young graduates. Further research should include more formal jobs, like banking, NGOs, and teaching, and objectively select comparable activities. Second, the DEA analysis does not allow very large samples, so generalization is restricted to the analyzed sector.

**Author Contributions:** Conceptualization, D.-M.A.N.; Data curation, D.-M.A.N.; Formal analysis, D.-M.A.N.; Funding acquisition, D.-M.A.N. and Z.B.; Investigation, D.-M.A.N.; Methodology, D.-M.A.N.; Supervision, P.-M.D.N.; Writing—original draft, D.-M.A.N.; Writing—review and editing, D.-M.A.N., B.A.A., D.B.K., V.K.B., J.M., Z.B. and P.-M.D.N. All authors have read and agreed to the published version of the manuscript.

**Funding:** This study was supported by the International Institute of Tropical Agriculture, under grant 2000001374 of the "Enhance Capacity to Apply Research Evidence (CARE) in Policy for Youth Engagement in Agribusiness and Rural Economic Activities in Africa", project number PJ 2459, funded by the International Fund for Agriculture Development (IFAD).

**Institutional Review Board statement:** The study was conducted according to the guidelines of the International Institute of Tropical Agriculture (IITA) Internal Review Board (IRB).

**Informed Consent Statement:** Informed consent was obtained from all participants involved in the study.

**Data Availability Statement:** The data used in this study are available upon request from the corresponding author.

**Acknowledgments:** The authors are grateful to the International Institute of Tropical Agriculture (IITA), which provided funding for this study, and the International Fund for Agriculture Development (IFAD) for funding the CARE Project. Finally, the authors would like to sincerely thank all anonymous persons for their very useful insights and helpful suggestions. The views expressed in this paper are those of the authors and should not be attributed to the organizations with which they are affiliated.

**Conflicts of Interest:** The authors declare no conflict of interest.

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
