# Peer review of "A Comparative Analysis of Technical Efficiency and Profitability of Agribusiness and Non-Agribusiness Enterprises in Eastern DRC"

_sustainability, doi:10.3390/su14148384_

Round 1

Reviewer 1 Report

This study refers to a very relevant area, such as the employability of young. In addition  the methodology used by the authors allows to obtain the results, which is rigorous and adequate. 

It is considered necessary for the topic include your relation with ODS (you could introduce these concept in the introduction).

Also you should add why the study is original and filling the gap in the literature. 

Finally, in reference to the conclusion, you need to say the necessity to continue investigating in this direction.

Author Response

Actually, all the raised comments were appreciative and we thank the reviewer. We addressed all the raised comments.

Reviewer 2 Report

I found the topic of this paper really exciting, but the authors must revise this paper before resubmitting it. There are several sections such as Introduction, methodology, and results that need to be improved in order to make readers understand the content. The authors needs to significantly improve the introduction section and section 2 regarding the description of the cassava community processing center model in order for it to be more line with the finding of their study.

Introduction:

  • In paragraph 3, the author states that agriculture will be less attractive to young people if the income they receive is not the same as other industries other than agribusiness such as trade, although the agribusiness sector itself also includes trade. Thus, I highly suggest that authors may provide further instances of industries other than agribusiness that are better suited to this study. It would be preferable if the examples of non-agribusiness industry utilized in the introduction were also connected to the industries identified in the result from Table 1, which were hairdressers, pharmacies, sewing workshops, restaurants and bistros, shops and magazines, transport, and mechanic. As a consequence, there will be coherence between the introduction and the outcomes.

Description of the cassava community processing center model

  • Why does the authors limit himself/herself to describing only about the Cassava Community Processing Center Model in Section 2? It would be great if the authors included information on the non-agribusiness industry that would serve as a comparative for this study. Thus, the author’s motivation for comparing the level of income earned in the agribusiness and non-agribusiness sectors (hairdressers, pharmacies, sewing workshops, restaurants and bistros, shops and magazines, transport, and mechanic) would be clearer and simpler to grasp for readers of this study.

Research Methodology:

  • Please provide a source in your table (Table 1).
  • In Section 3.3, which discusses analytical approaches of technical efficiency and profitability, the authors do not explain why Data Envelopment Analysis (DEA) should be used in this study. The authors should elaborate on why DEA was chosen to assess efficiency and profitability in this study, so that the reader understands the rationale for the selection.
  • However, since the authors describes the reason for using DEA in section 3.4. Thus, it is advised that sections 3.3 and 3.4 be integrated to make the article easier to follow.
  • Perhaps it is advisable to create formulae directly in Microsoft Word using the equation feature rather than copy and paste the images. This is because the letters in the equation are not the same size as the other letter formats, and certain elements of the picture are omitted on line 203. Additionally, if you copy equation images from another author’s work, you must mention the source; otherwise, you will be accused with plagiarism.

Results and discussion

  • Please give sources for all tables in the results and discussions section.
  • The authors simply attempted to interpret the number within the results in Section 4.3, which discusses the drivers of total technical efficiency. However, the authors made no attempt to explain the implications of the study findings in terms of real-world situations. It is anticipated that authors will also be able to infer the results in Table 5 and explain their implications.
  • There is a deficiency of explanation in the 4.4 section, which relates to the profitability of agricultural and non-agricultural enterprises. The findings in this part are very fascinating since it demonstrates that agribusiness creates more earnings than non-agribusiness. However, the authors just reported the findings without making any inferences. It is strongly recommended that authors provide the implications of their findings in Table 6 to bolster the study's assertions and explanations.

Conclusion and recommendations

  • Limitations to the study may be separated from the conclusions.

References

  • The authors must double check the recommended reference writing format from the sustainability journal because there are still some formatting errors.

Author Response

Actually, many aspects of the paper were appreciated by the reviewer and we took care of all the raised comments. We are grateful to the reviewer. 

Reviewer 3 Report

The paper is focused on the comparative assessment of agribusiness and non-agribusiness with regard to its technical efficiency and profitability. The methodology used for the comparison is Data Envelopment Analysis with its CRS and VRS models. The approach that is used is an appropriate one and commonly recognized, especially concerning the methodological choices. In general, the paper is well written, clear within its logic and synthetic. The text is well positioned versus existing literature concerning both the profitability and potential of agribusiness, especially in Eastern DCR, and methodology used for its assessment. The methodology is quite well presented and explained. The results are clearly presented and well interpreted. There are some concerns about the process, though, which needs to be clarified. First of all, the main purpose of using any DEA models is to assess comparable Decision Making Units. That assumption is especially important within the selection of variables, namely inputs and outputs, that are to be used for efficiency assessment. Since the comparison includes such DMUs as cassava processing centers and together with hairdressers and restaurants, the question arises whether this set of variables is appropriate for each of DMU type. Perhaps, the only solution would be to define different sets of variables (like agribusiness and non-agribusiness) and calculate efficiencies for every DMU within every set to give equal assessment conditions.

Secondly, the core of any DEA assessment is the set of input and outputs. These are not clearly presented in the text and the table with all the parameters is presented in the results section only. In my opinion, the variables should be presented earlier in the methodological section and should be discussed and perhaps some selection process should be applied in order to verify their significance in the process.

Finally, I cannot see clear message of the text concerning the profile of Sustainability journal, which leads to the conclusion that revied text should be re-written concerning this profile or submitted to other journal that would be more appropriate.

Some minor remarks are added to the attached manuscript.

Author Response

(The authors gave the same response as above.)

Reviewer 4 Report

Thank you for the opportunity to read the article.
The paper is interesting and well written.
However, several aspects need to be improved.
First of all, the aim of the research should be clarified and its practical application should be indicated. In addition, it is worth clarifying the hypotheses.
In conclusions, it is worth emphasizing the practical significance of the research results.

Author Response

(The authors gave the same response as above.)

Round 2

Reviewer 2 Report

The author should revise these more carefully and thoroughly.

  • Title: the current title is grammatically wrong.
  • Reference style. The second version used the wrong reference style. References are listed in the order of number. It means the very first reference should be numbered as 1. Line 30, the very first reference is numbered as 21. Why?
  • Introduction
  • Line 34, are there no more recent article or sources than 2014? Why the author used sources from 2014 and 2015 only?
  • Line 50-51, are there any source stating those careers are more profitable than agribusiness in the DRC? Or author may want to provide a reason those industries are chosen instead of others.
  • Line 55: please consider revise the phrase “agribusiness can be the best choice for young people.” This phrase sounds more subjective rather than objective. All the sources cited, none of them showed that” agribusiness can be the best choice for young people”
  • Materials and Methods
    • Please revise the format of all the formulars. Currently, there are visible line around the formulars.
  • References
  • Most references used in this manuscript are more than 5 years old. This will affect the up-to-date condition of this research.
  • Response to author
    • It is recommended that when author responds to reviewer’s comments, it’s better to address the responses in a complete sentence to avoid misunderstand. For example:

“There is a deficiency of explanation in the 4.4 section, which relates to the profitability of agricultural and non-agricultural enterprises. The findings in this part are very fascinating since it demonstrates that agribusiness creates more earnings than non-agribusiness. However, the authors just reported the findings without making any inferences. It is strongly recommended that authors provide the implications of their findings in Table 6 to bolster the study's assertions and explanations.” Reply: An explanation and inferences.

What do you mean by “An explanation and inferences”?

Author Response

Thanks again to the anonymous reviewer for his/her very useful comments. They were valuable to improve and develop this version of the manuscript. 

Reviewer 3 Report

The remarks included in the previous round has not been addressed extensively and therefore I'm bringing them out here:

  1. First of all, the main purpose of using any DEA models is to assess comparable Decision Making Units. That assumption is especially important within the selection of variables, namely inputs and outputs, that are to be used for efficiency assessment. Since the comparison includes such DMUs as cassava processing centers and together with hairdressers and restaurants, the question arises whether this set of variables is appropriate for each of DMU type. Perhaps, the only solution would be to define different sets of variables (like agribusiness and non-agribusiness) and calculate efficiencies for every DMU within every set to give equal assessment conditions. There is some justification within the answer of Authors but it is not enough and it does not change the text. The discussion on the condition of comparing such a different DMUs should be a starting point for the analysis.
  2. Secondly, the core of any DEA assessment is the set of input and outputs. These are not clearly presented in the text and the table with all the parameters is presented in the results section only. In my opinion, the variables should be presented earlier in the methodological section and should be discussed and perhaps some selection process should be applied in order to verify their significance in the process. The set of variables should be the first issue you address, comment and verify after presenting the assumptions of DEA modelling.
  3. Finally, I cannot see clear message of the text concerning the profile of Sustainability journal, which leads to the conclusion that reviewed text should be re-written concerning this profile or submitted to other journal that would be more appropriate. Authors refer to that in their answer but as for editorial requirements and not the scope of the journal. Therefore, this remark remains undressed in the text.

My conclusion remains the same as for previous round.

Author Response

(The authors gave the same response as above.)
